# Inhibition of Dipeptidyl Peptidase-4 Activates Autophagy to Promote Survival of Breast Cancer Cells via the mTOR/HIF-1α Pathway

**DOI:** 10.3390/cancers15184529

**Published:** 2023-09-12

**Authors:** Emi Kawakita, Fan Yang, Sen Shi, Yuta Takagaki, Daisuke Koya, Keizo Kanasaki

**Affiliations:** 1Department of Internal Medicine 1, Shimane University Faculty of Medicine, Izumo 693-8501, Shimane, Japan; 2Department of Diabetology & Endocrinology, Kanazawa Medical University, Uchinada 920-0293, Ishikawa, Japan; 3Department of Emergency Medicine, Affiliated Hospital of Southwest Medical University, Luzhou 646000, China; 4Division of Vascular Surgery, Affiliated Hospital of Southwest Medical University, Luzhou 646000, China; 5Division of Anticipatory Molecular Food Science and Technology, Medical Research Institute, Kanazawa Medical University, Uchinada 920-0293, Ishikawa, Japan

**Keywords:** DPP-4, autophagy, HIF-1α, metformin, apoptosis, breast cancer

## Abstract

**Simple Summary:**

The prevalence of certain cancers is high in the diabetic population. Dipeptidyl peptidase (DPP)-4, a therapeutic target for type 2 diabetes mellitus, is also involved in autophagic flux and cancer biology. In this paper, we reported that DPP-4 inhibition promotes breast cancer cell survival via induction of autophagy by the C-X-C motif chemokine 12/C-X-C receptor 4/mammalian target of rapamycin/hypoxia-inducible factor-1α axis. Metformin counteracts such an undesirable influence of DPP-4 inhibition on cancer biology. Our findings suggest that DPP-4 inhibitor, one of the most prescribed anti-diabetic drugs, could harm patients with certain cancers. The combination with metformin use could provide some clues to the use of DPP-4 inhibitors via the modulation of cancer autophagy-dependent survival signaling.

**Abstract:**

Autophagy plays a complex role in breast cancer cell survival, metastasis, and chemotherapeutic resistance. Dipeptidyl peptidase (DPP)-4, a therapeutic target for type 2 diabetes mellitus, is also involved in autophagic flux. The potential influence of DPP-4 suppression on cancer biology remains unknown. Here, we report that DPP-4 deficiency promotes breast cancer cell survival via the induction of autophagy by the C-X-C motif chemokine 12 (CXCL12)/C-X-C receptor 4 (CXCR4)/mammalian target of rapamycin (mTOR)/hypoxia inducible factor (HIF)-1α axis. DPP-4 knockdown and DPP-4 inhibitor KR62436 (KR) treatment both increased the levels of LC3II and HIF-1α in cultured human breast and mouse mammary cancer cells. The KR-induced autophagic phenotype in cancer cells was inhibited by treatment with the CXCR4 inhibitor AMD3100 and rapamycin. HIF-1α knockdown also suppressed breast cancer autophagy induced by KR. The autophagy inhibitor 3-methyladenine significantly blocked the KR-mediated suppression of cleaved caspase-3 levels and apoptosis in breast cancer cell lines. Finally, we found that the metformin-induced apoptosis of DPP-4-deficient 4T1 mammary cancer cells was associated with the suppression of autophagy. Our findings identify a novel role for DPP-4 inhibition in the promotion of breast cancer survival by inducing CXCL12/CXCR4/mTOR/HIF-1α axis-dependent autophagy. Metformin is a potential drug that counteracts the breast cancer cell survival system.

## 1. Introduction

Autophagy plays a complex role in tumor initiation, proliferation, and metastasis [1]. Cancers facilitate autophagy to survive microenvironmental stress and to increase growth and aggressiveness [2]. Breast cancer cells are more dependent on autophagy than normal epithelial cells, and this phenomenon is likely due to inherent deficiencies and to the increased metabolic and biosynthetic demands imposed by deregulated proliferation [3]. Mammalian target of rapamycin (mTOR) plays a crucial role in regulating autophagy [4]. mTOR exists in two distinct complexes, mTOR complex 1 (mTORC1) and mTOR complex 2 (mTORC2). Activation of mTORC1 by either nutrients or growth factors leads to the inhibition of autophagy via the phosphorylation of multiple autophagy-related proteins, which promote autophagic initiation and autophagosome nucleation [5]. mTOR inhibition strongly induces autophagy in yeast, even in the presence of nutrients [6]; however, the effect of mTOR inhibition on autophagic flux is cell type-dependent [7].

mTOR is believed to enhance the transcriptional activity of hypoxia-inducible factor (HIF)-1α [8], a transcription factor that plays a key role in hypoxia-induced autophagy [9]. HIF-1α is also a downstream target of mTOR in breast cancer cells [10]. Furthermore, the mTOR/HIF-1α axis is associated with breast cancer biology, including epithelial mesenchymal transition (EMT), invasion, metastasis, and resistance to chemotherapy [11].

Dipeptidyl peptidase (DPP)-4 inhibitors are second- or third-line drugs commonly used in the management of type 2 diabetes (T2DM) [12]. DPP-4 has been shown to affect multiple biological processes, such as autophagy [13], adhesion, immunomodulation, and apoptosis [14]. On the basis of the numerous biological functions of DPP-4, some preclinical studies have shown a potential link between DPP-4 inhibitors and cancer progression-related processes, including cancer growth [15], metastasis [16], and chemotherapeutic resistance [17]. In addition, current clinical research indicates that the use of DPP-4 inhibitors might be associated with an increased risk of cholangiocarcinoma [18]. Regarding the pleiotropic effects of DPP-4 inhibitors on cancer biology, we have previously shown that DPP-4 suppression accelerates breast cancer metastasis by inducing EMT through C-X-C motif chemokine 12 (CXCL12)/C-X-C receptor 4 (CXCR4)-mediated mTOR activation [19]. Furthermore, DPP-4 inhibition-induced autophagy can contribute to angiogenesis by enhancing the survival of endothelial progenitor cells [20].

Metformin is a widely prescribed hypoglycemic drug in T2DM patients [21] and is also a potent anticancer agent [22,23]. Clinical studies have also shown that metformin has a protective effect against cancer development in breast cancer patients with T2DM [24,25]. Mechanistically, the tumor suppressive effect of metformin is partly dependent on mTOR inhibition [23]. It has been reported that metformin also blocks HIF-1α accumulation through suppression of the mTOR pathway [26] in different types of cancer [27,28].

Here, we hypothesize that the CXCL12/CXCR4/mTOR/HIF-1α axis in DPP-4-deficient cells plays a key role in the activation of autophagy in breast cancer cells to promote survival. Additionally, we analyzed the influence of metformin on this autophagy-dependent cancer survival pathway.

## 2. Materials and Methods

### 2.1. Reagents and Antibodies

The DPP-4 inhibitor KR62436 hydrate (KR, K4264), CXCR4 inhibitor AMD3100 (AMD, A5602), rapamycin (R8781), and 3-methyladenine (3-MA, M9281) were purchased from Sigma (St. Louis, MO, USA). Metformin hydrochloride (metformin) was purchased from FUJIFILM Wako (136-18662, Osaka, Japan). The anti-β-actin antibody (A2228), anti-DPP-4 antibody (SAB2500328) and anti-phospho-p70 S6 Kinase (phosphorylated at The389) antibody (p-S6K, SAB4503952) were purchased from Sigma. The anti-HIF-1α (ab179483), anti-LC3 (ab192890), anti-CXCR4 (ab1670), and anti-CXCL12 (ab18919) antibodies were purchased from Abcam (Cambridge, UK). The anti-phospho-mTOR (p-mTOR, 2971), anti-mTOR (2983), anti-caspase-3 (9662), anti-cleaved caspase-3 (9661), anti-p70 S6 Kinase (S6K, 9202), anti-phospho-AKT (phosphorylated at Ser473) (p-AKT, 4058), and anti-AKT (9272) antibodies were purchased from Cell Signaling Technology (Danvers, MA, USA). The anti-p62 antibody (PM045) was purchased from MBL (Nagoya, Japan).

### 2.2. Cell Culture and Treatment

The experimental cell lines were obtained from the American Type Culture Collection (ATCC, Dallas, TX, USA). Human MCF 10A mammary epithelial cells (CRL-10317) were cultured in MEBM basal medium (CC-3151, Lonza, Alpharetta, GA, USA) supplemented with MEGM (CC-4136, Lonza). Human MCF7 breast cancer cells (HTB-22) were cultured in ATCC-formulated Eagle’s minimum essential medium (30-2003) with 10% fetal bovine serum (FBS). All cell lines above were grown at 37 °C in a 5% CO_2_ atmosphere. Human MDA-MB-231 breast cancer cells (HTB-26) were cultured in ATCC-formulated Leibovitz’s L-15 medium (30-2008) with 10% FBS. A mixture of CO_2_ and air is detrimental to MDA-MB-231 cells when using this medium for culture; therefore, MDA-MB-231 cells were cultured in a 0% CO_2_ atmosphere at 37 °C. Mouse 4T1 mammary cancer cells (CRL-2539) were cultured in RPMI-1640 medium (ATCC, 30-2001) with 10% FBS at 37 °C in a 5% CO_2_ atmosphere. These cells were used for up to 10 passages, and all analyses were performed within 6 months of obtaining the cells from ATCC. When the cells reached 70–80% confluence in 6-well plates, KR (50 μmol/L), AMD (30 μmol/L), 3-MA (5 mmol/L), rapamycin (1 μmol/L), or metformin (10 mmol/L) was added to the experimental medium for 48 h. The drug dose and exposure time were determined according to the parameters used in previous publications [19,29]. Cells were harvested for subsequent Western blot analyses.

### 2.3. Transfection Experiments

The 4T1 cell line was transfected with siRNA (100 nmol/L) targeting mouse DPP-4 (Invitrogen, Carlsbad, CA, USA). MCF 10A, MCF7, and MDA-MB-231 cells were transfected with siRNA (100 nmol/L) targeting human DPP-4 (Invitrogen) or siRNA (100 nmol/L) targeting human HIF-1α (Invitrogen). For the transfection experiments, all cell lines above were seeded at 1 × 10^5^ cells in 6-well plates. After the incubation overnight, the cells were washed and incubated with the serum-free Opti-MEM I medium (31985062, Thermo Fisher Scientific, Waltham, MA, USA). Then, the siRNA was transfected into the cells using with Lipofectamine RNAiMAX (13778-075, Invitrogen), according to the manufacturer’s instructions. After transfection, the cells were incubated for 24 h and subjected to analyses. For the HIF-1α siRNA transfections, the cells were treated with or without KR (50 μmol/L) for 48 h after transfection.

### 2.4. Western Blot Analysis

Proteins were harvested using RIPA lysis buffer containing phenylmethylsulfonylfluoride, protease inhibitor cocktail, and sodium orthovanadate (Santa Cruz Biotechnology, Santa Cruz, CA, USA) on ice. The amount of protein in each sample was quantified using a standard BCA assay kit (Thermo Fisher Scientific). The protein lysates were boiled in sodium dodecyl sulfate (SDS) sample buffer at 95 °C for 5 min, separated on SDS-polyacrylamide gels (5–20%), and then transferred onto PVDF membranes (Pall Corporation, Pensacola, FL, USA) using the semidry method. The membranes were blocked at room temperature (RT) with Tris-buffered saline (TBS) containing 0.05% Tween-20 (TBST) and 5% non-fat dry milk for 30 min and then incubated with antigen-specific primary antibodies at 4 °C overnight. The membranes were washed 3 times with TBST and then incubated with the corresponding peroxidase-conjugated secondary antibody at RT for 1 h. After 3 washes with TBST, the bands were detected by using an enhanced chemiluminescence detection system (34577, Thermo Fisher Scientific) and visualized using an ImageQuant LAS 400 camera system (GE Healthcare Life Sciences, Uppsala, Sweden). Immunoblotting was performed with primary antibodies against: AKT (1:2000), p-AKT (1:1000), caspase-3 (1:1000), cleaved caspase-3 (1:1000), CXCR4 (1:1000), CXCL12 (1:1000), HIF-1α (1:1000), LC3 (1:2000), mTOR (1:1000), p-mTOR (1:1000), S6K (1:1000), p-S6K (1:1000), p62 (1:1000), and β-actin (1:2500).

### 2.5. Mouse Breast Cancer Models

Cultured 4T1 cells (5 × 10^5^ cells in 20 μL of PBS) were injected into the mammary fat pad of each female BALB/c mouse (8 weeks old) (CLEA Japan, Inc., Tokyo, Japan). When the primary tumor size measured approximately 300 mm^3^ [29] or 500 mm^3^ [19], the mouse breast cancer model was treated with PBS (oral administration), KR (20 mg/kg/day, oral administration), AMD (7.5 mg/kg/day, intraperitoneal injection), metformin (200 mg/kg/day, intraperitoneal injection), KR and AMD, or KR and metformin. Seven days after treatment was initiated, the mice were sacrificed, and the primary tumor tissues and lungs were collected. The primary tumors were fixed in 4% paraformaldehyde for subsequent immunohistochemical staining. All animal experiments were approved by the IACUC of Kanazawa Medical University (protocol numbers 2014-89, 2014-101, 2018-16 and 2019-20). The details of the animal experimental protocols were previously described [19,29].

### 2.6. Immunohistochemistry (IHC)

Mouse tissue paraffin slides were deparaffinized in 2 changes of xylene for 5 min each and transferred to 100%, 95%, and 70% alcohol for 3 min each. After incubation in 10 mM citrate buffer (pH 6.0) for 30 min (microwave), the slides were incubated in 3% H_2_O_2_ at room temperature for 10 min to block endogenous peroxidase activity. The specimens were then blocked with 10% normal goat serum for 1 h. Next, the slides were incubated with appropriate primary antibody for 1 h at RT, washed with PBS, and incubated with the corresponding secondary antibody for 30 min at RT. The IHC mentioned above was performed using a Vectastain ABC rabbit IgG Kit (PK4001, Vector Laboratories, Burlingame, CA, USA). Then, the slides were incubated with DAB substrate solution (SK-4100, Vector Laboratories) and counterstained with hematoxylin. The slides were dehydrated through 4 changes of alcohol (75%, 95%, 100%, and 100%) for 5 min each and hyalinized in 3 changes of xylene. Primary antibodies against the following antigens were used for IHC staining: HIF-1α (1:500), LC3 (1:1000), and p62 (1:1000). Photographs at 400× magnification were obtained from five different areas of each sample with an ECLIPSE TSIF-APH microscope (NIKON, Tokyo, Japan). In the staining with anti-p62 antibody, the DAB staining area was evaluated by using NIH ImageJ software (ver. 1.52q).

For cell death analysis, mouse primary tumor sections were analyzed using the TUNEL Assay Kit HRP-DAB (ab206386, Abcam), according to the manufacturer’s instructions. Sections were digitalized using an ECLIPSE TSIF-APH microscope (NIKON), and the number of apoptosis cells was counted in three random areas (×400 magnification).

### 2.7. Fluorescence-Activated Cell Sorting (FACS)

Apoptosis was assayed using the Annexin V-FITC Apoptosis Staining/Detection Kit (ab14085, Abcam), according to the manufacturer’s instructions. MCF 10A, MDA-MB-231, and 4T1 cells were treated with PBS (control), KR (50 μM), and 3-MA (5 mM) for 48 h. The cells were collected (1 × 10^5^ cells), washed, and then resuspended in 500 μL of Annexin V binding buffer. Then, the cells were incubated with fluorescein isothiocyanate-labeled annexin V (FITC) and propidium iodide (PI) for 10 min in the dark. The cells in the early apoptosis process showing annexin V-positive and PI-negative staining were detected by using a Gallios Flow Cytometer (Beckman Coulter, Brea, CA, USA).

### 2.8. BrdU Cell Proliferation Assay

Cell proliferation was measured by using the BrdU Cell Proliferation ELISA Kit (ab126556, Abcam), according to the manufacturer’s instructions. The MCF 10A and MDA-MB-231 cells were each plated at 5 × 10^3^ cells in a 96-well plate and subsequently, cells were treated by KR (50 μmol/L), with or without AMD (30 μmol/L), metformin (10 mmol/L), or 3-MA (5 mmol/L), for 16 h. Then, BrdU was added to the wells, without assay background. After the incubation with BrdU for 3 h, the cells were fixed and then incubated with anti-BrdU detector antibody for 1 h at RT. Subsequently, the cells were washed and incubated with the peroxidase-conjugated secondary antibody for 30 min. Finally, the colored reaction was detected by using a Multimode Microplate Reader DTX880 (Beckman Coulter), with the absorbance set at a single wavelength of 450 nm (OD_450_). A high value of OD_450_ indicates a high rate of cell proliferation.

### 2.9. ATP/ADP Ratio Assay

The cellular ATP/ADP level was quantified by using the ADP/ATP Ratio Assay Kit (MAK135, Sigma), according to the manufacturer’s instructions. MCF 10A and MDA-MB-231 cells were cultured in 96-well white plates with clear bottoms at 5 × 10^3^ cells. After 24 h incubation with KR (50 μmol/L), KR with AMD (30 μmol/L), metformin (10 mmol/L), or 3-MA (5 mmol/L), the cells were lysed in nucleotide-releasing buffer. The ATP and ADP levels were determined by reading luminescence reaction (RLU) using a Multimode Microplate Reader CTX880 (Beckman Coulter).

### 2.10. Statistical Analysis

To assess significant differences, GraphPad Prism software (ver. 8.2.0) was used for the statistical analysis. The data are presented as the mean ± SEM. One-way ANOVA with Tukey’s multiple comparison test was used to determine significance, which was defined as a *p* value < 0.05.

## 3. Results

### 3.1. Breast Cancer Cells Were Highly Sensitive to DPP-4 Suppression-Induced Autophagy

To confirm the effects of DPP-4 deficiency on autophagy, we utilized siRNA to knock down DPP-4. Western blot analysis revealed that DPP-4 knockdown increased the levels of the autophagosome marker LC3II and decreased the expression of the autophagy substrate p62 in the human MCF7 and MDA-MB-231 breast cancer cell lines and mouse 4T1 mammary cancer cells (Figure 1A), suggesting autophagic induction by DPP-4 suppression. Interestingly, the induction of autophagy by DPP-4 knockdown was not observed in MCF 10A human normal breast epithelial cells (Figure 1A). Next, we analyzed the key molecules associated with DPP-4 inhibition-induced breast cancer metastasis in our previous study [19]. DPP-4 knockdown significantly promoted the levels of CXCL12, CXCR4, phospho-mTOR, and HIF-1α in breast cancer cell lines (Figure 1B). However, these molecules showed no alterations in the DPP-4 knockdown normal breast epithelial MCF 10A cells (Figure 1B).

### 3.2. CXCR4 Played an Essential Role in DPP-4 Inhibition-Induced Autophagy in Breast Cancer Cells

Next, we investigated the role of CXCR4 in autophagy induced by DPP-4 suppression. Treatment with the DPP-4 inhibitor KR62436 (KR) increased CXCR4, mTOR phosphorylation, HIF-1α, and LC3II, and decreased the levels of p62 in the human MCF7 and MDA-MB-231 breast cancer cell lines (Figure 2A); however, cotreatment with the CXCR4 inhibitor AMD3100 (AMD) abolished KR-induced phenotypes, such as mTOR phosphorylation, HIF-1α accumulation, and autophagic activation (Figure 2A). Similar data and trends were also obtained in the mouse 4T1 mammary tumor cell line (Figure 2B). There were no alterations in either KR-treated or CXCR4-inhibited normal breast epithelial MCF 10A cells (Figure 2A). Immunohistochemical (IHC) analysis in 4T1 primary tumors indicated that the KR-treated mice exhibited higher levels of LC3 and HIF-1α accumulation; AMD blocked these KR-induced alterations (Figure 2C).

### 3.3. Autophagic Induction by DPP-4 Inhibition Was Dependent on the mTOR/HIF-1α Axis

We next clarified whether DPP-4 inhibition-induced autophagy was mTOR dependent. Western blot analysis revealed that KR treatment of MCF7 and MDA-MB-231 cancer cell lines increased HIF-1α accumulation and autophagy compared to the control, while the mTOR inhibitor rapamycin dramatically attenuated DPP-4 deficiency-induced HIF-1α accumulation and autophagic activation, evaluated by LC3II/I ratio and p62 levels (Figure 3A). KR treatment alone did not influence either HIF-1α accumulation or autophagy levels in normal breast epithelial MCF 10A cells; rapamycin intervention significantly increased autophagy in the MCF 10A cells, even in the presence of a DPP-4 inhibitor (Figure 3A). Next, we confirmed the role of HIF-1α in autophagy regulated by DPP-4 inhibition. KR treatment-induced autophagy was significantly diminished in the HIF-1α siRNA-transduced MCF7 and MDA-MB-231 cells (Figure 3B). In the HIF-1α siRNA-treated MCF 10A cells, again, there were no alterations in the autophagic response (Figure 3B).

### 3.4. DPP-4 Inhibition Promoted Breast Cancer Cell Survival via Autophagy

Next, we determined the effects of DPP-4 inhibition-induced autophagy on apoptosis. Western blotting showed that DPP-4 inhibition decreased the levels of cleaved caspase-3, an apoptosis-associated protein, in the human MCF7 and MDA-MB-231 breast cancer cell lines; however, the autophagy inhibitor 3-methyladenine (3-MA) remarkably restored the levels of cleaved caspase-3 (Figure 4A). Similar trends were also observed in the mouse 4T1 mammary cancer cell line (Figure 4A). As expected, neither KR nor 3-MA treatment affected cleaved caspase-3 levels in normal breast epithelial MCF 10A cells (Figure 4A). We also performed an FACS analysis of cell viability and apoptosis to detect the cells in the early apoptotic process. The DPP-4 inhibitor KR decreased the ratio of cells with annexin-V positive and propidium iodide (PI) negative, indicating the early apoptotic cells in the 4T1 cells (Figure 4B), but not significantly in the MDA-MB-231 cells. Compared with KR treatment alone, cotreatment with KR and 3-MA significantly increased the proportion of early apoptotic cells in both human breast and murine mammary cancer cells (Figure 4B). There were no effects of KR and/or 3-MA on cell viability and apoptosis in normal MCF10A cells (Figure 4B).

Next, we evaluated the effect of DPP-4 suppression on the cell proliferation rate in human normal epithelial and breast cancer cells. The DPP-4 inhibitor treatment significantly increased the cell proliferation detected by BrdU incorporation in MDA-MB-231 cancer cells, but not in MCF 10A normal cells. Even in the presence of KR, the autophagy inhibitor 3-MA mitigated cell proliferation in both normal breast and breast cancer cells (Appendix A). CXCR4 inhibition by AMD3100 also canceled the KR-induced cell proliferation in breast cancer cells (Appendix A). In these cells, we also examined the changes in the ATP/ADP ratio to clarify cell viability. The ATP levels were increased by KR compared to the control, resulting in an increase ratio of ATP to ADP in the MDA-MB-231 cells. This KR-mediated ATP production was impaired by autophagy inhibition (Appendix A). As expected, there were no effects of the DPP-4 inhibitor and the autophagy inhibitor in the MCF 10A cells (Appendix A).

### 3.5. Metformin Reversed DPP-4 Deficiency-Induced Autophagy and Survival via Suppression of the mTOR/HIF-1α Axis in Mammary Tumors

Finally, we investigated whether the suppression of autophagic induction was involved in the antigrowth effects of metformin in DPP-4-deficient cancer cells [29], similar to the effect of the mTOR inhibitor rapamycin. Rapamycin has been known to specifically bind to the mTORC1 complex, but not mTORC2. First, we confirmed that DPP-4 inhibition by KR activated S6 Kinase (S6K), an mTORC1 downstream effector, and that rapamycin significantly suppressed the induction of phosphorylated S6K by KR in 4T1 murine mammary tumor cells (Appendix A). The level of mTORC2 downstream AKT phosphorylation was not altered by KR treatment (Appendix A), indicating that DPP-4 suppression targets the mTORC1 pathway rather than mTORC2 in breast cancer cells. Metformin treatment is well known to activate AMP-activated protein kinase (AMPK), an upstream negative regulator of mTORC1. In 4T1 cells, metformin significantly reduced phosphorylated mTOR, mTORC1 downstream S6K phosphorylation, HIF-1α, and LC3II, and increased p62, with or without KR, in vitro (Figure 5A, Appendix A). Metformin increased the ratio of AKT phosphorylation to total AKT, different from the effect of rapamycin in 4T1 (Appendix A).

Our recent study showed that metformin mitigates DPP-4 suppression-induced breast cancer growth and metastasis by mTOR regulation [29]. To assess the effect of metformin on DPP-4 inhibitor-induced breast cancer autophagy and survival, we performed IHC staining in 4T1 primary tumors. DPP-4 inhibitor KR treatment revealed a significant decrease in the p62-positive area (Figure 5B) and apoptotic tumor cells (Figure 5C) compared to that of the control tumor. Treatment with metformin increased p62 expression and the number of apoptotic cells, even under DPP-4 suppression (Figure 5B,C). However, metformin treatment alone did not impact basal p62 or apoptotic levels in 4T1 primary tumors (Figure 5B,C). Metformin also canceled the KR-induced cell proliferative effect in cultured MDA-MB-231 breast cancer cells, similar to the autophagy inhibitor 3-MA (Appendix A). Taken together, these results indicate that metformin can attenuate DPP-4 deficiency-induced breast cancer autophagy and survival through suppression of the mTOR/HIF-1α pathway (Figure 5D).

## 4. Discussion

We have provided clear evidence suggesting that DPP-4 inhibition could facilitate cancer growth, metastasis, and chemoresistance via the accumulation of CXCL12, the endogenous substrate of the DPP-4 enzyme [17,19,29]. Subsequently, relevant research has reported elsewhere that the DPP-4 inhibitor lessens the disease-free survival in patients with T2DM after curative resection for colorectal cancer [30]. In the current report, we focused on the pathological importance of DPP-4 deficiency in breast cancer autophagy. Here, we report the following: (1) DPP-4 knockdown induced autophagy and HIF-1α accumulation in a CXCL12/CXCR4/mTOR-dependent manner in human breast and mouse mammary cancer cells, but not in MCF 10A human normal breast epithelial cells; (2) a DPP-4 inhibitor suppressed the apoptosis of breast cancer cells via the induction of autophagy; (3) both rapamycin and metformin inhibited autophagy in cultured breast/mammary cancer cells, either with or without DPP-4 deficiency; and (4) in mouse mammary cancer cells, metformin mitigated DPP-4 deficiency-induced autophagy and promoted apoptosis through mTOR suppression. These data indicate a novel mechanism by which DPP-4 deficiency induces differential responses in autophagy and apoptosis between cancer cells and normal cells. Moreover, the antidiabetic drug metformin, which exerts a potent anticancer effect, combats the unfavorable effects of DPP-4 inhibitors by regulating the mTOR/HIF-1α axis.

Tumor cells activate autophagy in response to cellular stress, including hypoxia and the increased metabolic demands associated with rapid cell proliferation [31]. Autophagy-related stress tolerance can enable cell survival by maintaining energy production that can lead to tumor growth and therapeutic resistance [32]. Here, we found that breast cancer cells exhibited high sensitivity to DPP-4 deficiency-induced autophagy via the CXCL12/CXCR4-mediated pathway; however, such molecular mechanisms were not found in normal breast epithelial cell lines. Gavilan et al. also reported a difference in the autophagic response between normal breast cells and breast cancer cells [33]. MCF7 human breast cancer cells are more dependent on autophagy to maintain cell homeostasis under stress conditions, whereas nontumor MCF 10A cells are more dependent on proteasomes than autophagy [33]. Indeed, elevated autophagic activity has been observed in many cancers, including breast cancers [34].

Autophagic induction is associated with cancer cell survival. It was also reported that the high expression of the autophagosome marker LC3B is correlated with tumor progression and poor outcome in triple-negative breast cancer patients [35]. In our analysis, the inhibition of autophagy by 3-MA induced apoptosis in breast cancer cells. Furthermore, DPP-4 deficiency-augmented autophagy is crucial for the survival advantage in breast cancer cells treated with a DPP-4 inhibitor. Again, such autophagy-dependent cell survival, either with or without DPP-4 inhibition, was not found in normal breast epithelial cells. These observations suggest that the autophagic activation by DPP-4 deficiency might play different roles in cancer or noncancer cells. In fact, some preclinical studies revealed the favorable effects of DPP-4 inhibition-mediated autophagic activation in nontumor cells. Treatment with the DPP-4 inhibitor vildagliptin improved the survival rate after acute myocardial infarction by restoring the autophagic response in OLETF rats, a model of T2DM [36]. Another report showed that sitagliptin, a DPP-4 inhibitor, attenuated cadmium-induced testicular impairment and apoptosis via mTOR suppression-mediated autophagic activation [37]. However, the influence of DPP-4 inhibition-induced autophagy on cancer cell progression-related processes has not been elucidated. To our knowledge, this is the first report indicating that DPP-4 suppression affects the autophagic cell protection system in breast cancer cells. DPP-4 displays numerous biological functions in tumor and nontumor cells. Therefore, further research will be needed in this field.

The CXCL12/CXCR4 signaling pathway plays an important role in the growth and angiogenesis of primary breast cancer [38,39]. Correia et al. recently showed that CXCR4 expression is required for breast cancer proliferation and metastasis [40]. It was also reported that high CXCR4 expression is correlated with poor clinical prognosis in breast cancer patients [41]. Moreover, CXCL12-CXCR4 interactions were shown to stimulate the autocrine secretion of CXCL12 and induce cancer invasion and metastasis in CXCR4-positive esophageal cancers [42]. As expected, we also found that the aggressive breast cancer cell lines MDA-MB-231 and 4T1 exhibited higher CXCL12/CXCR4 levels than did MCF 10A normal epithelial breast cells. DPP-4 suppression augmented CXCL12/CXCR4 levels, which led to autophagy and HIF-1α in breast cancer cells; a CXCR4 inhibitor reversed these DPP-4 inhibition-induced alterations. CXCL12/CXCR4 signaling has also been shown to increase autophagic activity and decrease cytarabine-induced apoptosis in acute myeloid leukemia cells [43]. These findings suggest that the higher endogenous CXCL12/CXCR4 levels in breast cancer cells could determine the stronger response to DPP-4 deficiency-induced autophagy and apoptosis.

We have previously shown that the DPP-4 inhibitor KR induces expression of genes associated with mTOR pathway, particularly autophagy regulators (*Bnip3* and *Adm*), in mouse mammary primary tumor [29]. The CXCR4 inhibitor treatment decreased the KR-induced gene expression profile and breast cancer progression when compared to that of the KR alone group, suggesting that mTOR pathway activation is relevant for DPP-4 inhibitor-mediated breast cancer progression. mTORC1 is a well-known master regulator of autophagy [5]. In fact, the results we present here indicated that KR induces the phosphorylation of S6K in 4T1 cancer cells; rapamycin, an mTORC1 inhibitor, attenuates the autophagy induction by KR with the suppression of mTOR/S6K signaling. Moreover, we have reported that a DPP-4 inhibitor and shRNA-mediated DPP-4 knockdown both increases the expression of phosphorylated S6K in the primary tumors of 4T1 mammary tumor-bearing mice; metformin significantly repressed such a DPP-4 deficiency-induced mTORC1 pathway in vivo as well [29]. These results support the current data indicating that mTORC1 pathway activation is essential in breast cancer autophagy induction by DPP-4 inhibition.

HIF-1α is an important target in cancer therapy. The regulation of HIF-1α plays a crucial role in breast cancer progression and metastasis [44]. Hu et al. found that HIF-1α-induced autophagy promotes cell survival and resistance to antiangiogenic therapy in glioblastoma [45]. In addition, previous studies demonstrated that mTOR activation enhances the level of HIF-1α-mediated transcription [46]. The current data also indicated that mTOR/HIF-1α is a dominant pathway for DPP-4 inhibition-induced autophagy in breast cancer cells, but not in normal breast epithelial cells.

Finally, we found that metformin inhibited the DPP-4 deficiency-stimulated mTOR/HIF-1α/autophagy axis and induced breast cancer apoptosis. In addition to its hypoglycemic effect, metformin has been recognized as an anticancer agent. Metformin use was reported to be associated with a decreased risk of several cancers [47,48], including breast cancer [25,49,50]. A recent study indicated the antimetastatic effect of metformin on ovarian cancer by inhibiting mTOR-mediated HIF-1α activation [27]. Metformin also prevented pancreatic cancer progression via suppression of both mTOR and autophagic induction [51]. These findings suggest that mTOR inhibition by metformin is a relevant therapeutic strategy in breast cancer cells, showing higher HIF-1α and autophagic levels. However, our in vivo analysis revealed that metformin treatment alone had no impact on basal autophagy levels or apoptotic cell numbers in primary mammary specimens. These results were potentially due to the difference in metformin concentration in vitro and in vivo, which has been noted elsewhere [52]. Alternatively, metformin may preferentially target cancer cells with aberrant mTOR activation induced by the CXCL12/CXCR4-mediated pathway. Regarding this hypothesis, we have shown that metformin failed to suppress basal tumor growth in 4T1-bearing mice, but suppressed both growth and phosphorylated-mTOR/S6K levels in 4T1 mammary tumors, but only when co-administered with a DPP-4 inhibitor [29]. Despite the anticancer potential of this drug, some clinical trials showed no beneficial effect of metformin treatment on breast cancer patients [53,54,55]. In the future, further investigation is needed to elucidate the suitable conditions under which metformin can exert antitumor effects.

Altogether, DPP-4 inhibition could influence cell survival via the augmentation of autophagy in breast cancer cells. Physicians need to realize that DPP-4 has various functions in the cancer progression processes, both favorable and unfavorable.

## 5. Conclusions

In conclusion, we show here that DPP-4 inhibition accelerates breast cancer cell survival by inducing autophagy through CXCL12/CXCR4-mediated mTOR/HIF-1α activation; metformin abrogates such alterations induced by DPP-4 suppression. DPP-4 inhibition may induce differential responses in autophagy and cell survival between breast cancer cells and normal cells.

## Figures and Tables

**Figure 1 cancers-15-04529-f001:**
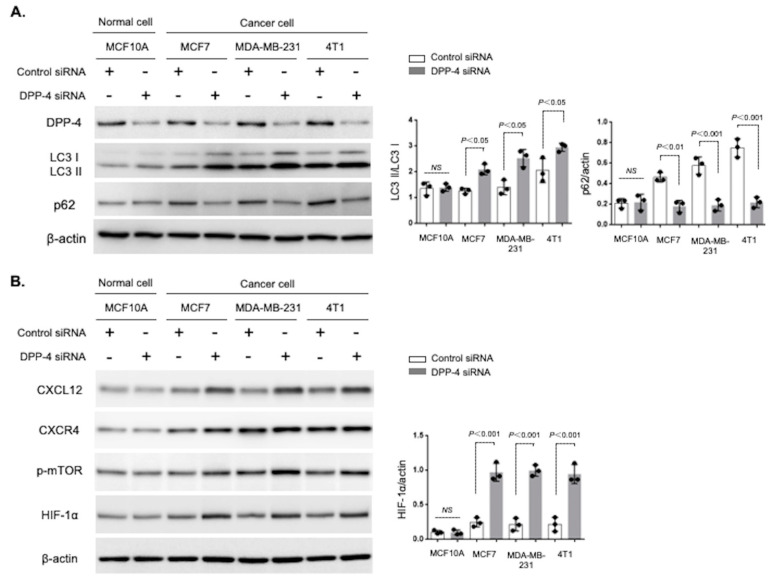
Silencing DPP-4 promotes autophagy and HIF-1α in breast cancer cells. (**A**). Western blot analysis of DPP-4, LC3, and p62 in MCF 10A, MCF7, MDA-MB-231, and 4T1 cells treated for 48 h in the presence or absence of DPP-4 siRNA (100 nM). Densitometric analysis of p62 levels normalized to β-actin levels and LC3II levels normalized to LC3I levels (*n* = 3 per group). (**B**). Western blot analysis of CXCL12, CXCR4, p-mTOR, and HIF-1α in MCF 10A, MCF7, MDA-MB-231, and 4T1 cells, treated with or without DPP-4 siRNA (100 nM). Densitometric analysis of HIF-1α levels relative to β-actin levels (*n* = 3 per group). The data in the graph are presented as the mean ± SEM. NS, not significant.

**Figure 2 cancers-15-04529-f002:**
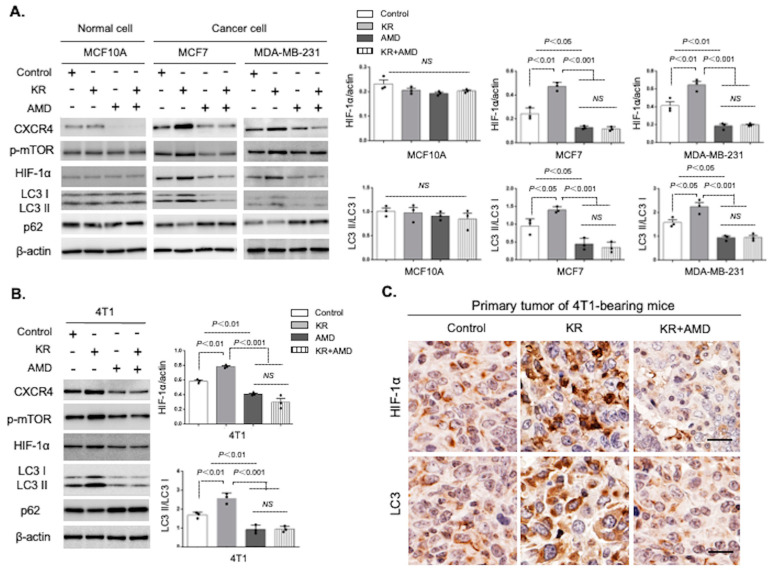
CXCR4 plays a critical role in DPP-4 inhibition-induced autophagy in human and mouse breast cancer cells. (**A**,**B**). The levels of CXCR4, p-mTOR, HIF-1α, LC3, and p62 were estimated by Western blot assays. Human MCF 10A, MCF7, and MDA-MB-231 cells (**A**) and mouse 4T1 (**B**) cells were treated with KR62436 (KR, 50 μM) and/or AMD3100 (AMD, 30 μM) for 48 h. Densitometric analysis of protein levels relative to β-actin or LC3I levels (*n* = 3 per group). The data in the graph are presented as the mean ± SEM. NS, not significant. (**C**). Immunohistochemistry analysis of HIF-1α and LC3 in 4T1 mammary tumor tissues following PBS (control), KR (20 mg/kg/day, oral injection), or KR and AMD (7.5 mg/kg/day, intraperitoneal injection) treatment for 7 days. Scale bars, 50 μm.

**Figure 3 cancers-15-04529-f003:**
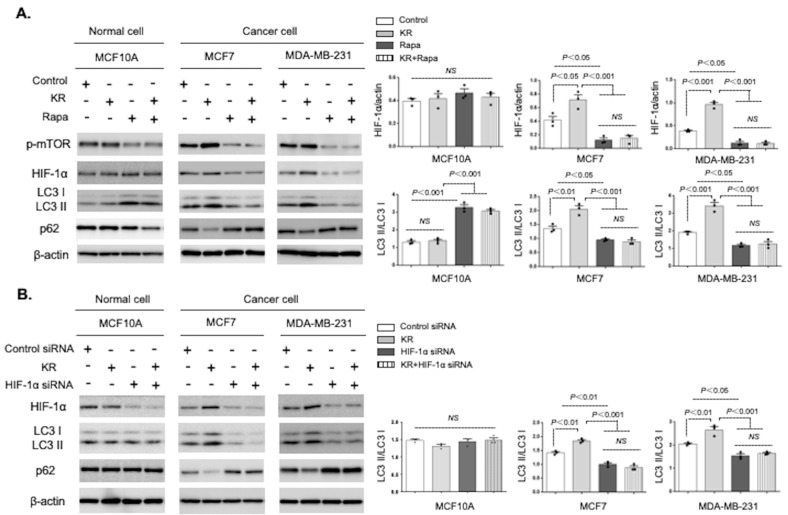
DPP-4 inhibition-induced autophagy is dependent on the mTOR/HIF-1α axis. (**A**). Western blot analysis of p-mTOR, HIF-1α, LC3, and p62 in MCF 10A, MCF7, and MDA-MB-231 cells treated with KR62436 (KR, 50 μM) and/or rapamycin (Rapa, 1 μM) for 48 h. Densitometric analysis of protein levels relative to β-actin or LC3I levels (*n* = 3 per group). (**B**). The levels of HIF-1α, LC3, and p62 were estimated by Western blot assays. MCF 10A, MCF7, and MDA-MB-231 cells were treated with KR (50 μM), HIF-1α siRNA (100 nM), or KR and HIF-1α siRNA for 48 h. Densitometric analysis of protein levels relative to β-actin or LC3I levels (*n* = 3 per group). The data in the graph are presented as the mean ± SEM. NS, not significant.

**Figure 4 cancers-15-04529-f004:**
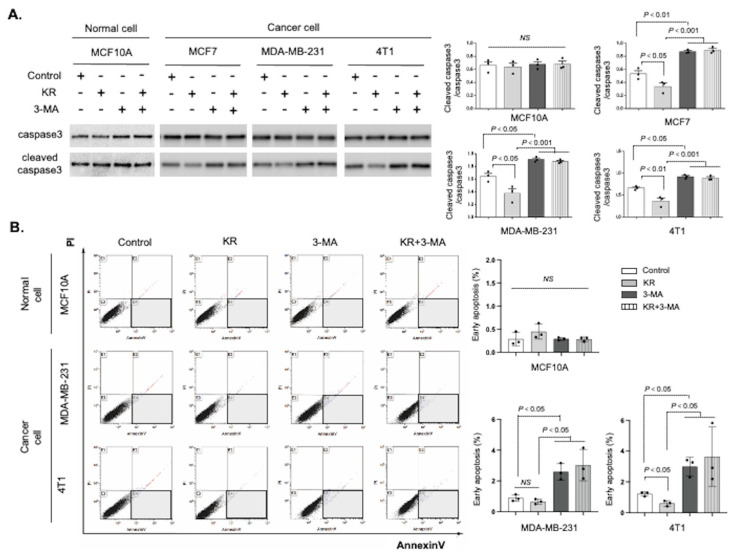
DPP-4 suppression promotes breast cancer survival by autophagic induction. (**A**). Western blot analysis of caspase-3 and cleaved caspase-3 in MCF 10A, MCF7, MDA-MB-231, and 4T1 cells treated with KR62436 (KR, 50 μM), 3-methyladenine (3-MA, 5 mM), or KR and 3-MA for 48 h. Densitometric analysis of cleaved caspase-3 levels relative to caspase-3 levels (*n* = 3 per group). (**B**). Rate of early apoptosis as indicated by staining in MCF 10A, MDA-MB-231, and 4T1 cells treated with KR (50 μM) and/or 3-MA (5 mM) for 48 h. The rates of early apoptosis were evaluated using annexin V and propidium iodide (PI) staining and flow cytometry. The cells labeled with annexin V+ and PI−, which are shown in the flamed area, were detected as the cells in the early apoptotic process (*n* = 3 per group). The data in the graph are presented as the mean ± SEM. NS, not significant.

**Figure 5 cancers-15-04529-f005:**
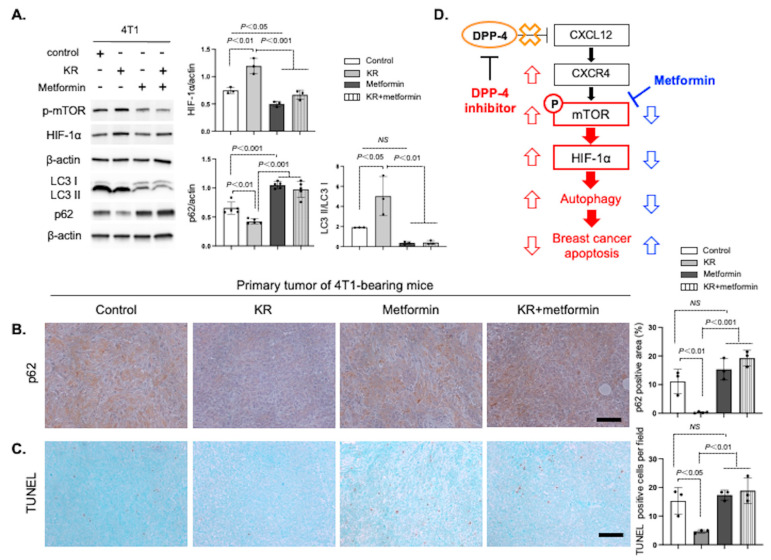
Metformin suppresses the DPP-4 inhibition-induced mTOR/HIF-1α/autophagy axis in 4T1 mammary tumors. (**A**). Western blot analysis of p-mTOR, HIF-1α, LC3, and p62 in 4T1 cells treated with KR62436 (KR, 50 μM), and/or metformin (10 mM) for 48 h. Densitometric analysis of protein levels relative to β-actin or LC3I levels (*n* = 3–5 per group). (**B**,**C**). Immunohistochemistry of primary mammary tumors for p62 (**B**) and apoptotic marker TUNEL staining (**C**) following PBS (control), KR (20 mg/kg/day, oral injection), metformin (200 mg/kg/day, intraperitoneal injection), or KR and metformin treatment for 7 days (*n* = 3 each group). Scale bars, 50 μm. Error bars represent the mean ± SEM. NS, not significant. (**D**) Schematic model of how metformin mitigates DPP-4 inhibitor-induced breast cancer cell survival.

## Data Availability

All data supporting the conclusions are available from the corresponding author upon request.

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
