# Peer review of "Inhibition of Dipeptidyl Peptidase-4 Activates Autophagy to Promote Survival of Breast Cancer Cells via the mTOR/HIF-1α Pathway"

_cancers, 2023, doi:10.3390/cancers15184529_

Round 1

Reviewer 1 Report

The study evaluated “Inhibition of dipeptidyl peptidase-4 activates autophagy to promote survival of breast cancer cells via the mTOR/HIF-1α pathway”  and designed very well.

This manuscript was prepared well with all parts of the manuscript and discussed with the results of the study very deeply and detail. 

Finally, the great article can be accepted to be published in this journal with the present form.

Author Response

We thank the reviewer for the positive comments. We also highly appreciate reviewer’s effort for spending the valuable time for us.

-------------------------------------------------------------------------------------------------------------------------

The study evaluated “Inhibition of dipeptidyl peptidase-4 activates autophagy to promote survival of breast cancer cells via the mTOR/HIF-1α pathway” and designed very well.

This manuscript was prepared well with all parts of the manuscript and discussed with the results of the study very deeply and detail. 

Finally, the great article can be accepted to be published in this journal with the present form.

We appreciate the positive comments. We have added some data to supplementary figure and some discussion in the revised manuscript according to the reviewer’s comments. Also, we carefully re-edited the whole manuscript to make it acceptable for journal publications.

Reviewer 2 Report

The manuscript entitled “Inhibition of dipeptidyl peptidase-4 activates autophagy to promote survival of breast cancer cells via 2 the mTOR/HIF-1α pathway” by Kawakita et al explored the impact of dipeptidyl 14 peptidase (DPP)-4 inhibitors, commonly prescribed for type 2 diabetes mellitus, on breast cancer cells. The authors reported that DPP-4 inhibition promotes breast cancer cell survival by inducing autophagy through the C-X-C motif chemokine 12 /C-X-C receptor 4 /mammalian target of rapamycin /hypoxia-inducible factor-1α axis. However, the authors show that this effect was countered by the use of metformin, a commonly prescribed medication for diabetes. The study suggests that the use of DPP-4 inhibitors may have harmful effects on patients with certain cancers, and combination therapy with metformin could provide a way to modulate cancer autophagy-dependent survival signaling.

Comments:

1.      The methods section needs to be improved. Authors have missed many information.

2.      It is not mentioned if the authors focused on mTORC1 or mTORC2 pathway. Accordingly, additional western blots must be shown.

3.      It is important to show the proliferation rate of these different cells used in the study under various conditions (e.g.: KR or KR+AMD or KR+Rapa). This is relevant for most of the figures.

4.      Authors have used 4T1 cells and injected orthotopically into the fat pad to observe the tumor size under various treatment conditions. Two important results are missing: 1) Kaplan-Meier plot and 2) Tumor size.

5.      In Figure 4B, for apoptosis study using Annexin V-FITC, the gating looks slightly different for MCF10A cells compared to all other cells. Please fix this. It is important to also use a viability dye to ensure that you are analyzing only viable cells.

Author Response

The manuscript entitled “Inhibition of dipeptidyl peptidase-4 activates autophagy to promote survival of breast cancer cells via 2 the mTOR/HIF-1α pathway” by Kawakita et al explored the impact of dipeptidyl 14 peptidase (DPP)-4 inhibitors, commonly prescribed for type 2 diabetes mellitus, on breast cancer cells. The authors reported that DPP-4 inhibition promotes breast cancer cell survival by inducing autophagy through the C-X-C motif chemokine 12 /C-X-C receptor 4 /mammalian target of rapamycin /hypoxia-inducible factor-1α axis. However, the authors show that this effect was countered by the use of metformin, a commonly prescribed medication for diabetes. The study suggests that the use of DPP-4 inhibitors may have harmful effects on patients with certain cancers, and combination therapy with metformin could provide a way to modulate cancer autophagy-dependent survival signaling.

----- We thank the reviewer for the response and valuable suggestions in the form of comments. We also highly appreciate the reviewer’s effort for spending your valuable time for us. These suggestions as below are made to improve our work. We are providing point-to-point answer to the reviewer’s questions and comments.

Comments:

  1. The methods section needs to be improved. Authors have missed many information. 

----- We thank the reviewer for the comment and sincerely apologize the confusion. We carefully re-edited the whole methods sections.

  1. It is not mentioned if the authors focused on mTORC1 or mTORC2 pathway. Accordingly, additional western blots must be shown. 

 ----- We thank the reviewer for the intellectual comment. We examined the downstream target of mTORC1 and mTORC2 pathway in DPP-4 inhibitor KR-treated 4T1 cancer cells. Western blot analysis represented that KR increased a mTORC1 downstream effector S6 kinase (S6K) phosphorylation in 4T1 cells; both rapamycin and metformin suppressed the KR-induced increase of phosphorylated mTOR and S6K. The mTORC1 pathway is known as an important upstream regulator of autophagy (J Clin Invest. 2015;125(1):25-32). In addition, we previously reported that a DPP-4 inhibitor and shRNA-mediated DPP-4 knockdown both increased the expression of phosphorylated S6K in primary tumor of 4T1 breast cancer-bearing mice; metformin significantly repressed such DPP-4 suppression-induced mTORC1 pathway in vivo (Kawakita E et al. Mol Cancer Res 2021.19(1):61-73). These results also support the current data of mTORC1 pathway activation by DPP-4 inhibitor in breast cancer cells.
     Regard with the mTORC2 pathway, we found that DPP-4 inhibition by KR exerted a trend of increase in AKT phosphorylation, a mTORC2 downstream target protein, but this alteration in KR-treated cells was not statistically significant. Indeed, it was reported that CXCL12/CXCR4 pathway could activate mTORC2 pathway in endothelial cell (Angiogenesis. 2016;19(3):359-71), thus DPP-4 suppression potentially activates mTORC2 pathway as well as mTORC1 depending on cell types. However, in our current study, we think that mTORC2 activation is not essential for the induction of autophagy in DPP-4 inhibitor-treated cancer cell. The reason why is because we also found that metformin, which exhibited the suppressive effect on autophagy induction by KR, also significantly increased the ratio of p-AKT/AKT either with or without KR in 4T1 cell. The molecular role of mTORC2 in cell autophagy regulation has not been fully elucidated, thus further investigation is needed in future.
Taken together, we mainly focused on mTORC1 pathway in this manuscript.

We have added the western blot data in supplementary figure S2 and we also mentioned this point in the discussion part.

  1. It is important to show the proliferation rate of these different cells used in the study under various conditions (e.g.: KR or KR+AMD or KR+Rapa). This is relevant for most of the figures.

----- We thank the reviewer for the important comment. We explored the cell proliferation rate by using a BrdU assay in human normal epithelial MCF10A cell and breast cancer MDA-MB-231 cell. The DPP-4 inhibitor KR increased the cell proliferation measured by BrdU incorporation compared to control in breast cancer cells. The cell proliferative effect of KR was attenuated by a CXCR4 inhibitor AMD3100, a mTOR inhibitor metformin, and an autophagy inhibitor 3-MA. DPP-4 inhibition had no effects on cell proliferation rate in MCF10A cells. Unexpectedly, in MCF10A cells, the 3-MA decreased the cell proliferation rate in the presence of KR. Hence, autophagy is important for cell proliferation in both normal breast cell and breast cancer cell even if its dependency on autophagy is different between each type of cells.
     We also evaluated an ATP/ADP ratio to test the cell viability in KR-treated cells. The ratio of ATP to ADP was increased by the KR treatment in MDA-MB-231 cells, indicating increase of cell proliferation rate. Again, these KR-induced changes were mitigated by AMD, metformin, or 3-MA in breast cancer cells. As expected, there were no effects of DPP-4 inhibition on the ATP/ADP ratio in normal cells. These results suggest that DPP-4 suppression increased breast cancer cell viability via the CXCR4/mTOR-mediated autophagy induction.
In fact, several studies have shown that autophagy inhibition could suppress the breast cancer cell proliferation and apoptosis (Anticancer Agents Med Chem. 2021;21(3):355-364, Int J Radiat Oncol Biol Phys. 2021;110(4):1234-1247, Cell Physiol Biochem. 2017;43(5):1829-1840).

We now added these data in supplementary figure S1 and re-edited an applicable text in previous manuscript.

  1. Authors have used 4T1 cells and injected orthotopically into the fat pad to observe the tumor size under various treatment conditions. Two important results are missing: 1) Kaplan-Meier plot and 2) Tumor size.

 ----- We thank the reviewer for this important comment. The primary tumor samples using for IHC staining is originated from our previous analysis reported in Cancer Research 2019 and Molecular Cancer Research 2021. We have already showed the change of primary tumor size in each treatment group (control, KR, KR+AMD, metformin or KR+metformin group) in previous report, thus we presented only follow-up data in this manuscript. We hope the represented data as below helps.

Yang F, Kawakita E et al. Cancer Res, 2019. 79(4): p. 735-746.

Kawakita E et al. Mol Cancer Res, 2021. 19(1): p. 61-73.

In these reports, we found that DPP-4 inhibitor KR treatment significantly increased the primary tumor growth compared to control treatment in 4T1-bearing female mice. CXCR4 inhibition by AMD and mTOR inhibition by metformin both mitigated the KR-induced primary tumor growth, indicating that DPP-4 inhibition induced tumor progression in the CXCR4/mTOR pathway dependent manner. However, the underlying mechanism of CXCR4/mTOR-mediated primary tumor growth by DPP-4 inhibitor was unknown, thus we further explored the effect of DPP-4 inhibition on autophagic process as the mTOR downstream pathway in this study.

      Regard with the Kaplan-Meier plot, we didn’t perform the analysis of survival rate in the breast cancer-bearing mice treated with KR for the following reasons.
1) We set 7-days treatment term to see the effects of drug compounds on breast cancer metastasis.
2) We couldn’t set the death as an endpoint for the ethical issue because 4T1 is highly progressive cell lines.
There are few studies showing data of Kaplan-Meier plot associated with DPP-4. Hao et al indicated that low DPP-4 expression in tumor impaired a survival time compared to high DPP-4 expression in patients with hepatocellular carcinoma (P=0.016, low DPP4=48.3 months vs high DPP4=68.4 months) (Braz J Med Biol Res 2020;53(4):e9114). Retrospective analysis using SEER- and Medicare-linked database also showed that T2DM patients taking a DPP-4 inhibitor had a worse trend of overall survival with HR 1.07 (95%CI: 0.93-1.25, p=0.33) in the breast cancer cohort (Front Oncol 2020;10:405).

Further research investigating the influence of DPP-4 inhibition on breast cancer mortality is also needed in future.

  1. In Figure 4B, for apoptosis study using Annexin V-FITC, the gating looks slightly different for MCF10A cells compared to all other cells. Please fix this. It is important to also use a viability dye to ensure that you are analyzing only viable cells.

----- We thank the reviewer for the important comment and we sincerely apologize for the confusion. We also used propidium iodide (PI) with Annexin-V for cell staining to divide viable cells from necrotic cells in annexin V-positive cell population. After re-analysis of the FACS data including the PI staining, DPP-4 inhibitor KR significantly decreased the rate of cells in early apoptotic process (detected by staining with Annexin-V positive and PI-negative) only in 4T1 murine mammary cancer cells. This KR-induced cell survival in 4T1 was canceled by co-incubation with the autophagy inhibitor 3-MA. Similar results were obtained in MDA-MB-231 human breast cancer cells but the decrease rate of apoptotic cells in KR-treated cells was not significant compared to control cells. This result was possibly due to the lower rate of apoptotic cells in control cancer cells. In MCF10A normal epithelial cells, neither KR nor 3-MA had no effects on cell apoptotic levels as expected.
We now fixed the FACS data in Figure 4B and re-edited the manuscript in method and result parts.

Reviewer 3 Report

The authors address an important topic by pointing out that the incidence of some cancers is high in the diabetic population. By studying dipeptidyl peptidase (DPP)-4, they indicate its importance as a therapeutic target for type 2 diabetes, it is also involved in autophagy and cancer cell biology. Thanks to the applied research techniques, they identify a new role of DPP-4 inhibition in promoting breast cancer survival through the induction of CXCL12/CXCR4/mTOR/HIF-1α axis-dependent autophagy. In addition, they indicate that metformin is a potential drug that counteracts the survival system of breast cancer cells.

The authors emphasize that metformin is a commonly prescribed hypoglycaemic drug. Clinical studies have also shown that metformin has a protective effect against the development of cancer in patients with breast cancer. The authors indicate that the suppressive effect of metformin on cancer may be partially dependent from mTOR inhibition or HIF-1α-78 accumulation.

Of course, despite the anti-cancer potential of this drug, some clinical trials have shown no beneficial effect of metformin treatment in breast cancer patients so research is needed to elucidate these mechanisms.

Although the authors put forward complex research hypotheses, it would be good to specify the main research goal.

The research as part of the work is complex and based on modern research techniques, as well as using several cell lines, which is why they are worth recommending and publishing in "CANCERS", but it needs a few clarifications and some minor adjustments:

1.      Please specify the choice of doses of the tested drugs for the experiment, whether they were made on the basis of previous experiments or on the basis of literature.

2.      In the western blot, the membranes were blocked and incubated overnight with the primary antigen-specific antibody at 4°C and then with the appropriate peroxidase-conjugated secondary antibody for 1 hour at room temperature.

Did the authors also incubate with the primary antibody for 1 hour?

3.      Regardless of the “conclusion”, it would be useful for the reader to briefly summarize the findings of the research, which could be placed at the end of the discussion.

I bring this remark because there are many results and it would be worth summarizing them briefly

Author Response

The authors address an important topic by pointing out that the incidence of some cancers is high in the diabetic population. By studying dipeptidyl peptidase (DPP)-4, they indicate its importance as a therapeutic target for type 2 diabetes, it is also involved in autophagy and cancer cell biology. Thanks to the applied research techniques, they identify a new role of DPP-4 inhibition in promoting breast cancer survival through the induction of CXCL12/CXCR4/mTOR/HIF-1α axis-dependent autophagy. In addition, they indicate that metformin is a potential drug that counteracts the survival system of breast cancer cells.

The authors emphasize that metformin is a commonly prescribed hypoglycaemic drug. Clinical studies have also shown that metformin has a protective effect against the development of cancer in patients with breast cancer. The authors indicate that the suppressive effect of metformin on cancer may be partially dependent from mTOR inhibition or HIF-1α-78 accumulation.

Of course, despite the anti-cancer potential of this drug, some clinical trials have shown no beneficial effect of metformin treatment in breast cancer patients so research is needed to elucidate these mechanisms.

Although the authors put forward complex research hypotheses, it would be good to specify the main research goal.

The research as part of the work is complex and based on modern research techniques, as well as using several cell lines, which is why they are worth recommending and publishing in "CANCERS", but it needs a few clarifications and some minor adjustments:

 -------- We thank reviewer’s positive comments and important suggestions for our manuscript. We also appreciate the reviewer’s effort for spending their valuable time for us. Now we addressed all the points raised from the reviewer.

  1. Please specify the choice of doses of the tested drugs for the experiment, whether they were made on the basis of previous experiments or on the basis of literature.

 ----- We thank the reviewer for the important comment. We determined the drug dose based on our previous report and past publications examining effect of each drugs on cancer progression. For the KR, AMD, rapamycin and metformin, we used same concentration as our previous research (Cancer research. 2019;79(4):735-746, Int J Mol Sci. 2020;21(3):805, and Mol Cancer Res. 2021;19(1):61-73). Regard with the KR, the concentration more than 10 μM has been shown to almost completely abolish DPP-4 catalytic activity similar to the inhibitory effects of DPP-4 inhibitors in clinic.
3-methyladenine is widely used at a concentration of 5 mM for utilizing as an autophagy inhibitor.

We now specified that the experimental dose was determined according to our previous publications in method part.

  1. In the western blot, the membranes were blocked and incubated overnight with the primary antigen-specific antibody at 4°C and then with the appropriate peroxidase-conjugated secondary antibody for 1 hour at room temperature.

 Did the authors also incubate with the primary antibody for 1 hour?

  ----- We sincerely apologize for the confusion. In the western blot analysis, the membranes were incubated with the primary antibody at 4°C overnight (around 16 h) and then incubated with the secondary antibody at room temperature for 1 h.
We now re-edited the method part and clearly described the incubation time in each step.

  1. Regardless of the “conclusion”, it would be useful for the reader to briefly summarize the findings of the research, which could be placed at the end of the discussion.

I bring this remark because there are many results and it would be worth summarizing them briefly

 ----- We thank the reviewer for the important suggestions. We now added simple summarization of main implication in the end of discussion part.